

# Association of heart rate trajectories with the risk of adverse outcomes in a community-based cohort in Taiwan

Cheng-Chun Wei[1,2], Pei-Chun Chen[3], Hsiu-Ching Hsu[4], Ta-Chen Su[4], Hung-Ju Lin[4], Ming-Fong Chen[4,5], Yuan-Teh Lee[4] and Kuo-Liong Chien[2,4]

[1] Division of Cardiology, Department of Internal Medicine, Shin Kong Wu Ho Su Memorial Hospital, Taipei, Taiwan

[2] Institute of Epidemiology & Preventive Medicine, College of Public School, National Taiwan University, Taipei, Taiwan

[3] Department of Public Health, China Medical University, Taichung, Taiwan

[4] Department of Internal Medicine, National Taiwan University Hospital, Taipei, Taiwan

[5] Cardiovascular Research Laboratory, Cardiovascular Center, Clinical Outcome Research and Training Center, Big Data Center, China Medical University Hospital, Taichung, Taiwan

Corresponding author
Kuo-Liong Chien,
klchien@ntu.edu.tw

## ABSTRACT

Heart rate trajectory patterns integrate information regarding multiple heart rate measurements and their changes with time. Different heart rate patterns may exist in one population, and these are associated with different outcomes. Our study investigated the association of adverse outcomes with heart rate trajectory patterns. This was a prospective cohort study based on the Chin-Shan Community Cardiovascular Cohort in Taiwan. A total of 3,015 Chinese community residents aged > 35 years were enrolled in a prospective investigation of cardiovascular risk factors and outcomes from 1990 to 2013. The primary outcome was all-cause mortality, and the secondary outcome was a composite of coronary artery disease and cerebrovascular accidents. The following trajectory patterns were identified: stable, 61%; decreased, 5%; mildly increased, 32%; and markedly increased, 2%. During follow-up (median, 13.9 years), 557 participants died and 217 experienced secondary outcomes. The adjusted hazard ratios of primary and secondary outcomes for participants with a markedly increased trajectory pattern were 1.80 (95% CI [1.18–2.76]) and 1.45 (95% CI [0.67–3.12]), respectively, compared to those for participants with a stable trajectory pattern. A markedly increased heart rate trajectory pattern may be associated with all-cause mortality risks. Heart rate trajectory patterns demonstrated the utility of repeated heart rate measurements for risk assessment.

## INTRODUCTION

Resting heart rate is a straightforward and easily obtainable clinical variable. Epidemiologic studies have demonstrated that elevated heart rate is a sensitive indicator of short life expectancies and is associated with an increased risk of adverse outcomes, such as all-cause mortality, cardiovascular diseases, diabetes, cancer, and stroke (*Kannel et al., 1987*; *Aune*

*et al., 2017*). This association has been demonstrated in diverse populations, including healthy individuals (*Zhang, Shen & Qi, 2016*) and individuals with hypertension (*Paul et al., 2010*), established coronary artery disease (*Diaz et al., 2005*), and heart failure (*Chioncel et al., 2017*). However, some inconsistencies, such as the role of changes in the heart rate, require further clarification in a longitudinal study.

Most previous studies used a single-point heart rate measurement, such as the baseline heart rate, for analysis (*Jouven et al., 2009*; *Cooney et al., 2010*; *Tverdal, Hjellvik & Selmer, 2008*). The major drawback of this measurement is the possibility of dramatic changes in heart rate between examinations. Therefore, some researchers measured not only baseline heart rates but also differences in heart rates between examinations and found that heart rate differences or changes were more sensitive predictors than a single-point measurement (*Paul et al., 2010*; *Nauman et al., 2011*; *Vazir et al., 2018*). To study changes in heart rate with outcomes, covariates play an important role. Several covariates, such as age, body weight and metabolic factors, vary in the long-term follow-up and tremendously affect the results. Combining changes in heart rate with time-dependent covariates can help predict the results more accurately (*Vazir et al., 2018*). Moreover, different trajectory patterns of changes in heart rate can exist in a population with different outcomes (*Chen et al., 2017*).

To the best of our knowledge, few studies have investigated repeated heart rate measurements through heart rate trajectories combined with time-dependent covariates. This study aimed to identify different heart rate trajectory patterns using longitudinal heart rate data and to study the risk of adverse outcomes among different heart rate trajectory patterns.

## MATERIALS & METHODS

### Study design and participants

This prospective cohort study was based on the Chin-Shan Community Cardiovascular Cohort. The details of this cohort have been described previously (*Lee et al., 2000*). All patients provided written informed consent for publication of the information and personal medical history, and the study protocol conformed to the Declaration of Helsinki guidelines. The study was approved by the Institutional Research Board of National Taiwan University Hospital (approval number: 201003001R). Beginning in 1990, a total of 3,602 Chinese community residents aged >35 years in Northern Taiwan were enrolled in a prospective investigation of cardiovascular risk factors and outcomes. All-cause mortality and cardiovascular events were monitored until the end of 2013. We excluded participants who were using anti-hypertensive or anti-arrhythmia medications ($n = 391$); whose baseline and follow-up heart rate data were missing ($n = 162$); with key-in error ($n = 1$); whose heart rhythm demonstrated complete atrioventricular block or pacemaker rhythm ($n = 5$); and with atrial fibrillation ($n = 28$). After these exclusions, 3,015 participants were included in the study population. During the 1992–1993 follow-up period, 2,386 participants were available for heart rate analysis. Data were collected from 1,786 and 1,133 participants during the 1994–1995 and 2001–2002 follow-up periods, respectively. The flow diagram of cohort selection is shown in Fig. 1.

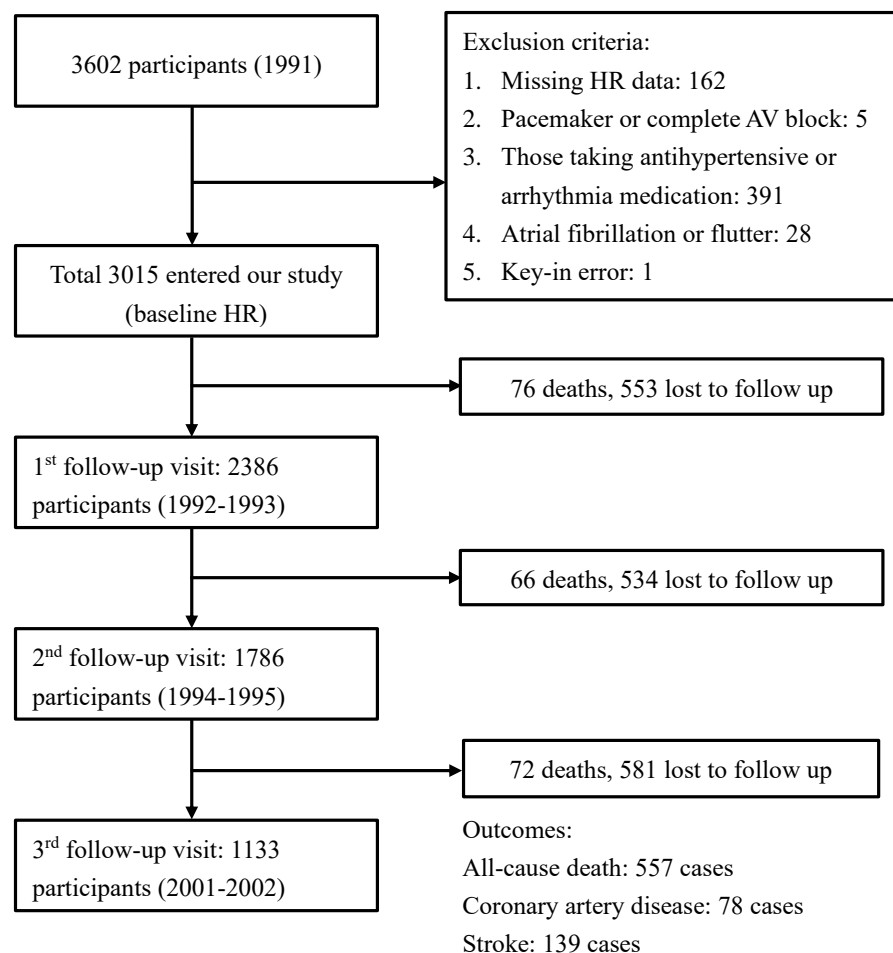

**Figure 1 Flow diagram of the cohort selection process.** The diagram shows the events, exclusion criteria, and the number of participants in each visit.

The participants' anthropometric and lifestyle data, medical history and current health conditions were assessed using interview questionnaires in 2-year cycles through the assistance of trained young faculty and medical students.

### Definitions of heart rate measurements

A 12-lead electrocardiography was performed in each participant to obtain heart rates, and two cardiologists blindly evaluated the results. Participants were instructed to avoid alcohol, food or drink with caffeine, and smoking 8 h before examination. Four repeated heart rate measurements were obtained during 1990 and 2002. The first heart rate reading was obtained during 1990–1991, and the follow-up heart rate readings were obtained during 1992–1993, 1994–1995, and 2001–2002. Heart rate parameters were the baseline and final heart rates. Baseline heart rate was defined as the heart rate measured at the first examination, and the final heart rate was defined as the last available heart rate during follow-up. The change in heart rate was defined as the change in heart rates

between examinations. When data were missing between examinations, we omitted the missing heart rate. For example, if the second examination was missed, then we calculated differences in the heart rates of the third and first examinations. Group-based trajectory models (GBTMs) of heart rates were based on statistical analyses, and individuals with similar patterns of changes in heart rate were grouped together. The heart rate coefficient of variation (CV) was defined as the ratio of the standard deviation to the average heart rate.

## Outcomes

The primary outcome was all-cause mortality, and the secondary outcome was a composite of coronary artery disease and cerebrovascular accidents (CVAs). Coronary artery disease was defined as nonfatal myocardial infarction, fatal coronary artery disease, or hospitalization for percutaneous coronary intervention or coronary artery bypass surgery, including hospitalization for angina, and coronary angiography showing >50% luminal stenosis. Fatal coronary artery disease was considered to have occurred if hospital records confirmed fatal myocardial infarction or if coronary heart disease was listed on the death certificate as the underlying and most plausible cause of death. CVAs were defined as sudden neurological deficits of the vascular origin that lasted for more than 24 h. Primary and secondary outcomes identified during follow-up visits were reviewed by physicians on the Committee of Mortality and Morbidity within the study team. Four cardiologists reviewed the completed interview questionnaires, medical records, and laboratory reports to determine whether each event met the protocol established by the project steering committee. The study team invited all participants to have biennial follow-up visits and examinations. Relatives of non-respondents were contacted to obtain information on the health status of non-responsive individuals. Medical students were responsible for administering the questionnaires and interviewing participants. We reviewed the hospitalization and outpatient service records corresponding to the questionnaires. Moreover, we regularly collected death certificate documents from the local health station and performed oral interviews with relatives and family members about death events to confirm the causes of death.

## Definitions of potential risk factors

Baseline characteristics of participants, including personal medical history, smoking habits, alcohol habits, lifestyle choices, exercise frequency, socioeconomic status, and family history, were collected through questionnaires. Physical examinations were conducted by trained physicians. Laboratory tests were conducted after a 12-hour fasting period. Blood pressure was measured after 10 min of rest using a mercury sphygmomanometer. This measurement was obtained with participants in a comfortable seated position, with their arms supported and positioned at the level of the heart. Blood pressure was measured twice in the right arm. If the readings varied by more than 10 mmHg, then an additional reading was performed. The average of these blood pressure measurements was used as the final blood pressure.

Body mass index (BMI) was calculated as weight (kg)/height$^2$ (m$^2$). Diabetes mellitus (DM) was defined as fasting glucose $\geq$126 mg/dL or as the use of oral hypoglycemic
or insulin medication. The estimated glomerular filtration rate (eGFR) was calculated using the Cockcroft–Gault formula (*Davies et al., 2018*; *O'Hare et al., 2007*). An eGFR <60 mL/min/1.73 m² indicated chronic kidney disease. Individuals were defined as smokers or never smokers; the smokers group included current smokers and those who had quit smoking. Regular exercise was defined as at least 20 min of physical activity two or three times per week.

## Statistical analysis

For descriptive analyses, continuous variables are described as mean (± standard deviation), and categorical variables as numbers and percentages. Characteristics of demographic data across groups were compared using an analysis of variance for continuous variables and using chi-square test for categorical variables of sex, history of hypertension, DM, atrial fibrillation, smoking, and sports.

The GBTM analysis was performed using SAS software (version 9.4; SAS Institute, Cary, NC) with the Traj package to identify trajectory patterns of long-term changes in heart rate. We adopted the censored normal model because heart rate is a continuous variable. We initially assumed the trajectory patterns to be cubic and treated heart rate as a dependent variable and time as an independent variable. A repeated trajectory analysis was performed by changing the number of groups from two to five. The Bayesian information criterion was used to estimate the number of trajectory patterns. The number of groups with the highest Bayesian information criterion was considered the appropriate one. After the number of groups was decided, we further tested each group of trajectories as linear, quadratic, or cubic to confirm the accurate graphical shape of change in heart rate by selecting the highest polynomial order to best characterize each trajectory group (*Nagin, 1999*; *Nagin & Odgers, 2010*). To test the robustness of our GBTM, we also analyzed trajectory patterns of only three repeated heart rate measurements obtained during 1990–1991, 1992–1993, and 1994–1995 and compared them with those of four repeated heart rate measurements.

Baseline and final heart rates were treated as both categorical and continuous variables in this study. To analyze the baseline and final heart rates as categorical variables, they were evaluated according to quartile. The change in heart rate was considered a continuous variable by calculating differences in heart rates between the first follow-up examination and baseline, between the second and first follow-up examinations, and between the third and second follow-up examinations. The heart rate CV was considered a continuous variable.

A survival curve was calculated using the Kaplan–Meier method. For Cox regression analyses, adjusted hazard ratios were reported with models 1 and 2. Model 1 was adjusted with age and sex only. Model 2 was adjusted with all covariates. We used baseline covariates to adjust for the baseline heart rate, GBTM of the heart rate, and CV of the heart rate, and we used time-dependent covariates to adjust for the final heart rate and change in heart rate. These covariates are established predictors of outcomes. Baseline covariates included age, sex, systolic blood pressure (SBP), diastolic blood pressure (DBP), diabetes status, smoking, cholesterol, BMI, and eGFR <60 mL/min/1.73 m². The time-dependent covariates were age, BMI, cholesterol, diabetes status, SBP, and DBP (*Vazir et al., 2018*).

Statistical significance was defined as a two-tailed $P < 0.05$. All analyses were performed with SAS version 9.4 (SAS Institute).

## RESULTS

Tables 1 and 2 show baseline characteristics of participants based on the quartile of the baseline and final heart rates. No significant differences in baseline heart rate were found among the groups. For the final heart rate, group 1 included participants with heart rate <61 beats per minutes (bpm); group 2 comprised those with heart rate between 62 and 68 bpm; group 3 comprised those with heart rate between 69 and 75 bpm; and group 4 comprised those with heart rate >76 bpm. Participants in group 4 were likely to have the following characteristics: female sex; higher BMI, cholesterol, and blood pressure; smoker; and diabetes. The four cubic trajectories of resting heart rates derived from the GBTM are shown in Fig. 2. We labeled them according to the trajectory pattern: stable ($n = 1,828$; 61%), decreased ($n = 155$; 5%), mildly increased ($n = 958$; 32%), and markedly increased ($n = 74$; 2%). Participants with stable, mildly increased, and markedly increased heart rate patterns started with lower baseline heart rates, whereas those with a decreased pattern started with higher baseline heart rates, but the trend decreased later. The trend of an increased heart rate was the highest for the markedly increased pattern and the lowest for the stable pattern. Table 3 shows baseline characteristics of each group. Sex, BMI, and cholesterol were significantly different among the groups. Participants in the markedly increased trajectory pattern group were likely to have higher BMI, cholesterol, and DBP than those in the other groups. Participants in the decreased trajectory pattern group were likely to be women, have a history of hypertension, and have higher SBP, and those with stable trajectory pattern were likely to be men, be smokers, have lower BMI and cholesterol levels, and have lower SBP and DBP.

Kaplan–Meier survival curves for all-cause mortality as well as the composite of outcomes of coronary artery disease and CVA according to the four heart rate trajectory patterns are shown in Figs. 3 and 4. The markedly increased trajectory pattern showed the worst survival rate, and the log-rank test results for all-cause mortality among the four patterns of heart rate change were significant ($p = 0.011$). For the composite outcome of coronary artery disease and CVA, no significant differences were found among the four heart rate trajectory patterns ($p = 0.77$). The adjusted hazard ratios of primary and secondary outcomes for participants with markedly increased trajectory pattern were 1.80 (95% CI [1.18–2.76]; $p = 0.007$) and 1.45 (95% CI [0.67–3.12]; $p = 0.34$), respectively, compared with those with the stable trajectory pattern (Table 4). For GBTM of the heart rate with three repeated heart rate measurements, four heart rate trajectories were identified, and their trends were similar to the results of the four repeated heart rate measurements. The adjusted hazard ratios of the primary and secondary outcomes for participants with markedly increased trajectory pattern were 1.62 (95% CI [1.10–2.38]; $p = 0.015$) and 0.87 (95% CI [0.40–1.88]; $p = 0.71$) (Table 5), respectively.
**Table 1** Baseline characteristics of participants based by baseline heart rate quartile.

| Number (%) and Mean (SD) | Group 1 (Baseline HR ≤58) (n = 766) | Group 2 (59 ≤Baseline HR ≤64) (n = 762) | Group 3 (65 ≤Baseline HR ≤72) (n = 809) | Group 4 (Baseline HR ≥73) (n = 678) | P-value |
|---|---|---|---|---|---|
| Age (years) | 54.0 ± 11.9 | 53.7 ± 12.3 | 53.6 ± 12.3 | 54.3 ± 12.4 | 0.68 |
| Sex (female) | 407 (53.1%) | 396 (52.0%) | 416 (51.4%) | 355 (52.4%) | 0.92 |
| BMI (kg/m²) | 23.3 ± 3.5 | 23.3 ± 3.2 | 23.2 ± 3.3 | 23.2 ± 3.3 | 0.86 |
| Cholesterol (mg/dL) | 195.9 ± 46.3 | 197.7 ± 43.8 | 196.5 ± 42.7 | 195.9 ± 44.9 | 0.85 |
| DM | 98 (12.9%) | 73 (9.6%) | 102 (12.6%) | 85 (12.6%) | 0.16 |
| Hypertension | 150 (19.7%) | 168 (22.2%) | 155 (19.4%) | 142 (21.0%) | 0.51 |
| Smoking | 261 (34.1%) | 287 (37.7%) | 311 (38.4%) | 246 (36.3%) | 0.30 |
| Alcohol | 217 (28.3%) | 233 (30.6%) | 250 (30.9%) | 219 (32.3%) | 0.42 |
| Exercise | 119 (15.5%) | 109 (14.3%) | 104 (12.9%) | 96 (14.2%) | 0.51 |
| SBP (mmHg) | 121.6 ± 17.7 | 122.7 ± 18.4 | 122.0 ± 18.7 | 123.1 ± 17.6 | 0.36 |
| DBP (mmHg) | 75.7 ± 10.5 | 76.5 ± 10.6 | 75.4 ± 10.3 | 75.9 ± 9.8 | 0.22 |
| eGFR <60 mL/min | 172 (22.7%) | 162 (21.4%) | 176 (22.0%) | 165 (24.4%) | 0.55 |
| All-cause mortality | 141 (18.4%) | 136 (17.9%) | 151 (18.7%) | 129 (19%) | 0.95 |
| Coronary artery disease and cerebral vascular accident | 55 (7.2%) | 64 (8.4%) | 54 (6.7%) | 40 (5.9%) | 0.30 |

Notes.
Abbreviations: BMI, body mass index; DBP, diastolic blood pressure; DM, diabetes mellitus; eGFR, estimated glomerular filtration rate; HR, heart rate; SBP, systolic blood pressure; SD, standard deviation.

**Table 2** Baseline characteristics of participants by final heart rate quartile.

| Number (%) and Mean (SD) | Group 1 (Final HR ≤61) (n = 788) | Group 2 (62 ≤Final HR ≤68) (n = 791) | Group 3 (69 ≤Final HR ≤75) (n = 701) | Group 4 (Final HR ≥76) (n = 735) | P-value |
|---|---|---|---|---|---|
| Age (years) | 59.5 ± 12.2 | 58.5 ± 12.3 | 58.4 ± 12.0 | 59.7 ± 12.2 | 0.09 |
| Sex (female) | 346 (43.9%) | 415 (52.5%) | 399 (56.9%) | 414 (56.3%) | <0.001 |
| BMI (kg/m²) | 23.3 ± 3.2 | 23.6 ± 3.3 | 23.7 ± 3.7 | 24.0 ± 3.9 | 0.001 |
| Cholesterol (mg/dL) | 199.5 ± 43.5 | 198.1 ± 42.4 | 202.9 ± 43.9 | 203.5 ± 42.7 | 0.06 |
| DM | 95 (13.3%) | 93 (12.9%) | 116 (18.3%) | 164 (25.2%) | <0.001 |
| Hypertension | 126 (19.0%) | 121 (18.3%) | 142 (24.6%) | 164 (28.7%) | <0.001 |
| Smoking | 337 (42.8%) | 275 (34.8%) | 231 (33.0%) | 262 (35.7%) | <0.001 |
| Alcohol | 264 (33.5%) | 241 (30.5%) | 201 (28.7%) | 213 (29%) | 0.16 |
| Exercise | 119 (15.1%) | 117 (14.8%) | 95 (13.6%) | 97 (13.2%) | 0.66 |
| SBP (mmHg) | 123.6 ± 20.6 | 124.6 ± 19.5 | 126.5 ± 19.5 | 128.9 ± 19.7 | <0.001 |
| DBP (mmHg) | 74.5 ± 10.9 | 75.2 ± 11.4 | 77.2 ± 11.8 | 78.9 ± 11.7 | <0.001 |
| eGFR <60 mL/min | 193 (24.7%) | 161 (20.6%) | 154 (22.0%) | 167 (22.9%) | 0.26 |
| All-cause mortality | 144 (18.3%) | 139 (17.6%) | 118 (16.8%) | 156 (21.2%) | 0.15 |
| Coronary artery disease and cerebral vascular accident | 63 (8%) | 49 (6.2%) | 50 (7.1%) | 51 (6.9%) | 0.58 |

Notes.
Abbreviations: BMI, body mass index; DBP, diastolic blood pressure; DM, diabetes mellitus; eGFR, estimated glomerular filtration rate; HR, heart rate; SBP, systolic blood pressure; SD, standard deviation.

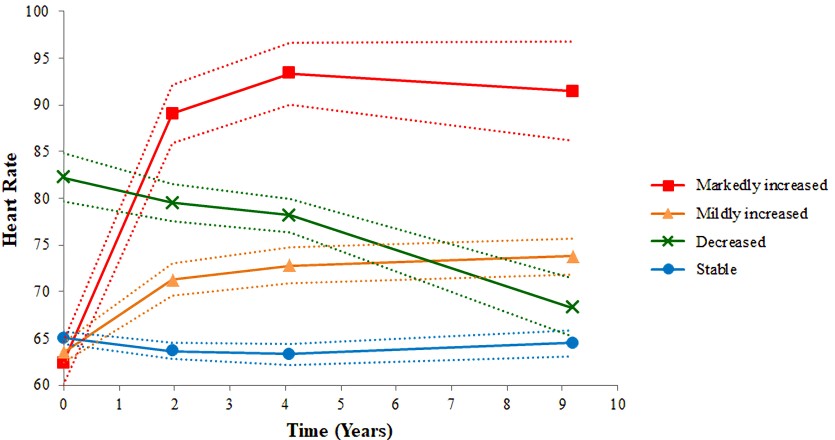

**Figure 2  Four heart rate trajectory patterns identified by the group-based trajectory model.** The figure shows four trajectory patterns in four different colors. The red line indicates markedly increased pattern. The orange line indicates mildly increased pattern. The green line indicates decreased pattern. The blue line indicates stable pattern. The dotted line is the 95% confidence interval of heart rate.

**Table 3  Baseline characteristics of participants based on the GBTM of the heart rate.**

| Number (%) and Mean (SD) | Stable (n = 1,828) | Decreased (n = 155) | Mildly increased (n = 958) | Markedly increased (n = 74) | P-value |
|---|---|---|---|---|---|
| Age (years) | 54 ± 12.1 | 55 ± 13.4 | 53.3 ± 12.2 | 55.3 ± 12.3 | 0.23 |
| Sex (female) | 880 (48.1%) | 92 (59.4%) | 559 (58.4%) | 43 (58.1%) | <0.001 |
| BMI (kg/m²) | 23 ± 3.1 | 23.1 ± 3.4 | 23.7 ± 3.6 | 24.1 ± 4.4 | <0.001 |
| Cholesterol (mg/dL) | 194.1 ± 44.1 | 199.4 ± 44.9 | 199.8 ± 44.5 | 207.6 ± 45.5 | 0.001 |
| DM | 197 (10.8%) | 18 (11.8%) | 130 (13.6%) | 13 (17.6%) | 0.08 |
| Hypertension | 348 (19.2%) | 44 (28.6%) | 205 (21.5%) | 18 (24.3%) | 0.025 |
| Smoking | 713 (39%) | 53 (34.2%) | 314 (32.8%) | 25 (33.8%) | 0.011 |
| Alcohol | 590 (32.3%) | 47 (30.3%) | 259 (27%) | 23 (31.1%) | 0.041 |
| Sport | 265 (14.5%) | 19 (12.3%) | 134 (14%) | 10 (13.5%) | 0.88 |
| SBP (mmHg) | 121.6 ± 18.5 | 126.8 ± 18.3 | 122.9 ± 17.2 | 125 ± 19.7 | 0.002 |
| DBP (mmHg) | 75.1 ± 10.2 | 77.1 ± 10.7 | 77 ± 10.2 | 78.8 ± 11.7 | <0.001 |
| eGFR <60 mL/min | 420 (23.2%) | 44 (28.6%) | 194 (20.3%) | 17 (23.3%) | 0.10 |
| All-cause mortality | 338 (18.5%) | 33 (21.3%) | 163 (17%) | 23 (31.1%) | 0.030 |
| Coronary artery disease and cerebral vascular accident | 127 (7%) | 10 (6.5%) | 69 (7.2%) | 7 (9.5%) | 0.86 |

**Notes.**
   Abbreviations: BMI, body mass index; DBP, diastolic blood pressure; DM, diabetes mellitus; eGFR, estimated glomerular filtration rate; HR, heart rate; SBP, systolic blood pressure; SD, standard deviation.

## DISCUSSION

### Main findings

The study demonstrated an association between heart rate trajectories and all-cause mortality, but no association was evident between heart rate trajectories and cardiovascular events. Our study was primarily based on the analysis of heart rate in sinus rhythm.

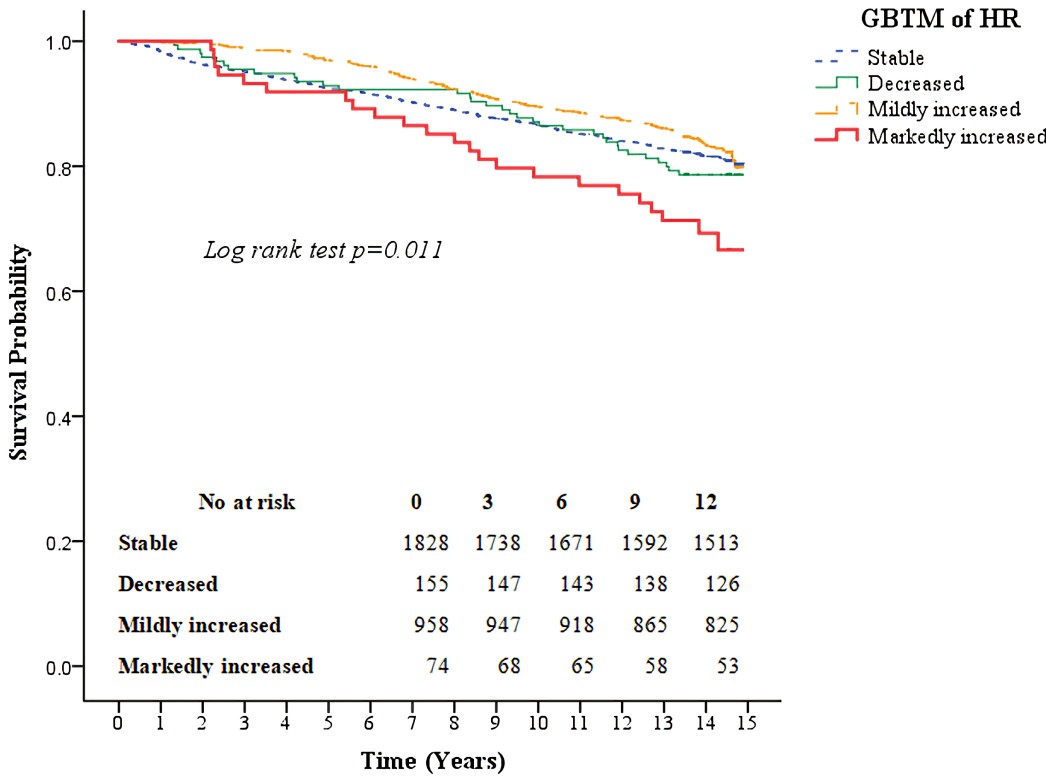

**Figure 3  Kaplan–Meier survival curves for all-cause mortality based on group-based trajectory model of the heart rate.** The red line indicates the survival curve of markedly increased pattern. The orange line indicates the survival curve of mildly increased pattern. The green line indicates the survival curve of decreased pattern. The blue line indicates the survival curve of stable pattern.

Participants with atrial fibrillation were excluded because previous literature showed that the impact of heart rate on adverse outcomes differed among patients with sinus rhythm and those with atrial fibrillation (*Kotecha et al., 2017*; *Laskey et al., 2015*). We also excluded participants using antihypertensive medications due to their strong influence on heart rate (*Whelton et al., 2017*; *McAlister et al., 2009*). Determining the risk of all-cause mortality from the baseline heart rate is sometimes arbitrary, particularly when the change in heart rate or variation with time is high in the cohort population. The baseline heart rate was not a good surrogate marker in this study. Based on the heart rate trajectories, only half of the participants had a stable heart rate, whereas the other half had a heart rate variation over 15% in one decade.

   Heart rate trajectories with time-dependent covariates adjustment will predict outcomes better in this situation. Participants with a decreased trajectory pattern started with the highest baseline heart rate among the groups. The risk of all-cause mortality in patients with a decreased trajectory pattern was not higher than that in patients with a stable trajectory pattern. Although participants with a decreased trajectory pattern started with higher baseline heart rates, the heart rate trajectory of the decreased pattern continued to decrease over time. The detrimental effect of a high baseline heart rate was diminished

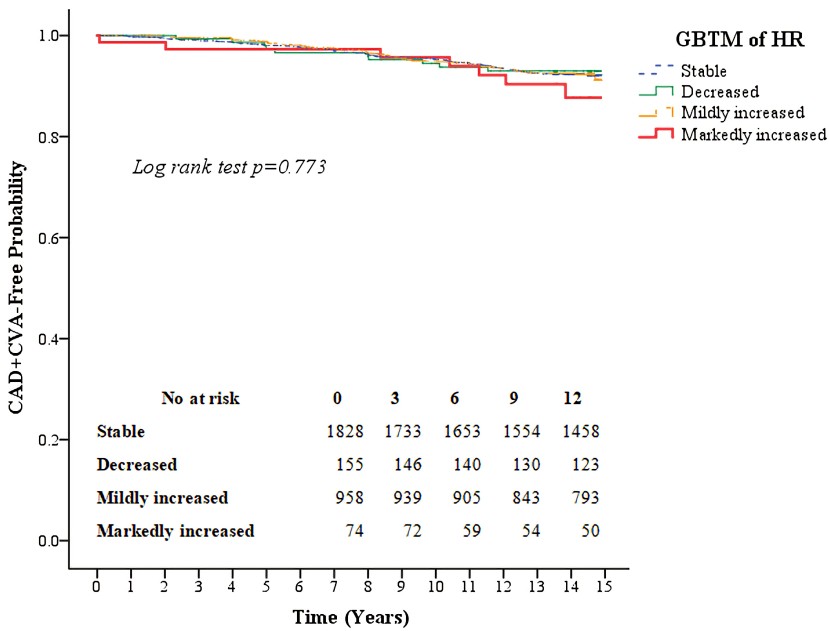

**Figure 4** **Kaplan–Meier survival curves for the composite outcome of coronary artery disease and cerebral vascular accidents based on the group-based trajectory model of the heart rate.** The red line indicates the survival curve of markedly increased pattern. The orange line indicates the survival curve of mildly increased pattern. The green line indicates the survival curve of decreased pattern. The blue line indicates the survival curve of stable pattern.

by the decrease in heart rate over time. However, participants with a stable trajectory pattern and those with markedly increased trajectory pattern started with similar baseline heart rates, but the heart rate trajectory of the markedly increased pattern increased over time. An approximate 50% increase in heart rate was found from the baseline in 2 years. The risk of all-cause mortality for those with markedly increased trajectory pattern was significantly higher than for those with stable trajectory pattern. The faster the heart rate increases, the higher the all-cause mortality rate. This finding confirmed that the heart rate trajectory could be a more sensitive predictor of adverse outcomes than a single heart rate measurement.

In our study, there was a gap between the last heart rate measurement and final outcomes. This temporal relationship demonstrates that the observed associations were not due to reverse causation. Further, we compared the results of GBTM for heart rate with three and four repeated heart rate measurements and found that the trajectories were very close and that the markedly increased trajectory pattern was associated with a significantly higher risk of all-cause mortality than the stable pattern in both models. This finding confirmed the reliability of our heart rate group classification and showed the possibility to predict long-term heart rate trajectory patterns with earlier repeated heart rate measurements.

Compared with the GBTM for heart rate, the change in heart rate did not correlate well with adverse outcomes. This finding may have been caused by the change in heart rate being considered as a continuous variable because it was difficult to predict a complex and

**Table 4  Hazard ratio and 95% confidence interval of the heart rate for adverse outcomes.**

| | Outcome | | | |
|---|---|---|---|---|
| | **All-cause mortality** | **P-value** | **Coronary artery disease and cerebral vascular accident** | **P-value** |
| Events (per 1000 person-years) | 14.36 | | 5.59 | |
| Adjusted hazard ratio (95% CI) | | | | |
| **Model 1** | | | | |
| Baseline HR (categorical) | | | | |
| (Group 2 vs. group 1) | 0.89 (0.70–1.13) | 0.33 | 1.09 (0.76–1.57) | 0.63 |
| (Group 3 vs. group 1) | 0.95 (0.75–1.19) | 0.63 | 0.88 (0.60–1.28) | 0.49 |
| (Group 4 vs. group 1) | 0.96 (0.75–1.22) | 0.72 | 0.76 (0.51–1.15) | 0.19 |
| Baseline HR (per 5 beats per minute) | 0.99 (0.96–1.04) | 0.85 | 0.95 (0.89–1.01) | 0.11 |
| Final HR (per 5 beats per minute) | 1.05 (1.02–1.09) | 0.006 | 1.00 (0.94–1.06) | 0.98 |
| Final HR (categorical) | | | | |
| (Group 2 vs. group 1) | 1.06 (0.84–1.34) | 0.61 | 0.86 (0.59–1.25) | 0.43 |
| (Group 3 vs. group 1) | 1.07 (0.84–1.37) | 0.59 | 1.04 (0.72–1.51) | 0.84 |
| (Group 4 vs. group 1) | 1.28 (1.02–1.60) | 0.035 | 0.96 (0.66–1.39) | 0.81 |
| ΔHR (per 5 beats per minute) | 1.02 (0.97–1.07) | 0.41 | 0.93 (0.86–1.00) | 0.033 |
| GBTM of HR | | | | |
| (Decreased vs. stable) | 1.06 (0.74–1.52) | 0.75 | 0.90 (0.47–1.72) | 0.75 |
| (Mildly increased vs. stable) | 0.94 (0.78–1.14) | 0.52 | 1.06 (0.79–1.43) | 0.69 |
| (Markedly increased vs. stable) | 1.86 (1.22–2.84) | 0.004 | 1.54 (0.72–3.30) | 0.27 |
| CV of HR | 1.97 (0.59–6.57) | 0.27 | 0.76 (0.12–5.05) | 0.78 |
| **Model 2** | | | | |
| Baseline HR (categorical) | | | | |
| (Group 2 vs. group 1) | 0.89 (0.70–1.13) | 0.32 | 1.01 (0.70–1.47) | 0.95 |
| (Group 3 vs. group 1) | 0.90 (0.71–1.14) | 0.38 | 0.82 (0.55–1.20) | 0.30 |
| (Group 4 vs. group 1) | 0.92 (0.72–1.17) | 0.50 | 0.74 (0.49–1.12) | 0.15 |
| Baseline HR (per 5 beats per minute) | 0.99 (0.95–1.03) | 0.65 | 0.95 (0.89–1.01) | 0.11 |
| Final HR (per 5 beats per minute) | 1.02 (0.98–1.06) | 0.47 | 0.97 (0.90–1.03) | 0.27 |
| Final HR (categorical) | | | | |
| (Group 2 vs. group 1) | 1.17 (0.91–1.51) | 0.22 | 0.94 (0.62–1.42) | 0.77 |
| (Group 3 vs. group 1) | 1.00 (0.77–1.31) | 0.98 | 0.95 (0.63–1.44) | 0.82 |
| (Group 4 vs. group 1) | 1.09 (0.85–1.41) | 0.51 | 0.81 (0.54–1.21) | 0.30 |
| ΔHR (per 5 beats per minute) | 1.02 (0.97–1.08) | 0.45 | 0.92 (0.85–1.00) | 0.05 |
| GBTM of HR | | | | |
| (Decreased vs. stable) | 1.02 (0.71–1.46) | 0.94 | 0.81 (0.42–1.55) | 0.52 |
| (Mildly increased vs. stable) | 0.92 (0.76–1.11) | 0.39 | 0.98 (0.72–1.32) | 0.87 |
| (Markedly increased vs. stable) | 1.80 (1.18–2.76) | 0.007 | 1.45 (0.67–3.12) | 0.34 |
| CV of HR | 1.88 (0.56–6.36) | 0.31 | 0.81 (0.11–5.72) | 0.83 |

**Notes.**

Abbreviations: CI, confidence interval; HR, heart rate; GBTM, group-based trajectory model; CV, coefficient of variance.

Model 1: Adjusted for age, sex.

Model 2: Adjusted for age, sex, systolic blood pressure, diastolic blood pressure, diabetes mellitus, smoking, alcohol, cholesterol, body mass index, estimated glomerular filtration rate <60 mL/min/1.73 m$^2$.

**Table 5** Hazard ratio and 95% confidence interval of the heart rate for adverse outcomes according to the heart rate trajectories based on repeated measurements of three heart rates.

| | Outcome | | | |
|---|---|---|---|---|
| | All-cause mortality | *P*-value | Coronary artery disease and cerebral vascular accident | *P*-value |
| **Model 1** | | | | |
| GBTM of HR | | | | |
| Decreased vs. stable | 1.13 (0.84–1.54) | 0.42 | 0.90 (0.52–1.57) | 0.71 |
| Mildly increased vs. stable | 0.92 (0.76–1.12) | 0.41 | 1.13 (0.84–1.51) | 0.42 |
| Markedly increased vs. stable | 1.73 (1.19–2.53) | 0.004 | 1.16 (0.54–2.48) | 0.71 |
| **Model 2** | | | | |
| GBTM of HR | | | | |
| Decreased vs. stable | 1.09 (0.80–1.48) | 0.59 | 0.85 (0.49–1.48) | 0.56 |
| Mildly increased vs. stable | 0.91 (0.75–1.10) | 0.34 | 1.00 (0.74–1.35) | 1.00 |
| Markedly increased vs. stable | 1.62 (1.10–2.38) | 0.015 | 0.87 (0.40–1.88) | 0.71 |

**Notes.**

Abbreviations: CI, confidence interval; HR, heart rate; GBTM, group-based trajectory model; CV, coefficient of variance.

Covariates: age, sex, systolic blood pressure, diastolic blood pressure, diabetes mellitus, smoking, alcohol, cholesterol, body mass index, estimated glomerular filtration rate <60 mL/min/1.73 m$^2$, and atrial fibrillation.

Model 1: Adjusted for age, sex.

Model 2: Adjusted for age, sex, systolic blood pressure, diastolic blood pressure, diabetes mellitus, smoking, alcohol, cholesterol, body mass index, estimated glomerular filtration rate <60 mL/min/1.73 m$^2$.

non-linear trend for the risk of heart rate outcomes. The heart rate CV showed the extent of variability in relation to the mean heart rate but lacked information regarding how the heart rate changed with time; therefore, it correlated poorly with outcomes.

For the secondary outcomes, there are several possible reasons for the insignificant results. First, a low events rate for secondary outcomes was responsible for the low statistical power in our population. Second, considering the effect size of trajectories on secondary outcomes, our sample size may have been too small for the analysis.

## Comparisons with previous studies

Previous literature emphasized the role of resting heart rate as a predictor of adverse outcomes, and resting heart rate was regarded as an indicator of general health (*Lindgren et al., 2018*). In one large cohort investigating the association of resting heart rate with cause-specific mortality, the authors found that heart rate over a decade was associated with a risk of death from cancer, cardiovascular diseases, and other causes. The hazard ratio of mortality was higher for cardiovascular diseases than for cancer. Our study used coronary artery diseases and stroke as secondary outcomes, which included mostly death from cardiovascular diseases. The impact of heart rate trajectory with outcomes was consistent with findings from a previous research; however, due to limited numbers of events, the results were not statistically significant (*Seviiri et al., 2018*).

Some studies reported that heart rate achieved during follow-up examinations was a better predictor of risk than the baseline heart rate (*Paul et al., 2010*; *Kotecha et al., 2017*). Other studies showed that the change in heart rate was positively correlated with all-cause mortality and non-fatal cardiovascular events (*Seviiri et al., 2018*; *Takahama et al., 2013*;

*Vazir et al., 2017*; *Kitai et al., 2017*). Clinical benefits were also reported to be associated with heart rate reductions (*McAlister et al., 2009*; *Swedberg et al., 2010*). In the present study, the heart rate trajectory consisted of not only the baseline heart rate or a single measurement of heart rate but also the change in heart rate over time. By recognizing the patterns of change in heart rate rather than the absolute numerical heart rate value, adverse outcomes can be predicted intuitively. Chen et al. found that heart rate trajectories could predict artery stiffness. Compared with that study, we followed our participants for years after establishing the heart rate trajectory model to demonstrate the relationship between trajectories and all-cause mortality (*Chen et al., 2017*).

Participants in the markedly increased heart rate pattern group were likely to have higher BMI, DBP, and cholesterol. Several studies showed the association of these risk factors with heart rate. Lower BMI is associated with cardiorespiratory fitness. Regular exercise can improve cardiorespiratory fitness, which was proven by some studies to decrease resting heart rate (*Quan et al., 2014*). Resting heart rate is an indicator of autonomic function. Autonomic dysfunction leads to a higher resting heart rate in hypertension, hyperlipidemia, and metabolic syndrome (*Inoue et al., 2007*; *Sun et al., 2012*). However, further studies are necessary to clarify the role of heart rate with these risk factors and adverse outcomes.

## Implications for clinical management

Our study was based on a community cohort, and most participants were healthy individuals. Although the external validity of our study was not tested, our results highlighted the possibility of investigating the heart rate not only using a single measurement but also using trajectory patterns. In addition, our study demonstrated the utility of repeated heart rate measurements for risk assessment.

Heart rate, as a component of vital signs, is underrated in the long term. The most widely accepted "normal heart rate" ranges from 60 to 100 bpm (*Fleming et al., 2011*). In the present study, most participants' heart rates were in the normal heart rate range; however, the markedly increased trajectory pattern was associated with a higher risk of all-cause mortality than the stable trajectory pattern. The importance of heart rate trajectory patterns cannot be overstressed. Because wearable devices have become increasingly popular, they may be helpful for recording large amounts of heart rate data, thereby allowing data analysis algorithms to define the trajectory more accurately and easily.

## Study strengths and limitations

This study had three major strengths. First, our study was based on a large sample size and included a long-term follow-up cohort study, both of which could have reduced the possibility of selection bias. Second, this study used several heart rate parameters, including the GBTM of heart rate, to estimate the effects of changes in heart rate on adverse outcomes. Third, our heart rate measurements were based on the ECG not the pulse rate. Participants would rest for 10 min before the heart rate was measured. This type of standardization enhanced the accuracy of heart rate measurements. This study also had limitations. Our results were based on the GBTM, and group identifications were not certain. The GBTM may identify additional groups to accommodate non-normality in the data rather than true

latent groups. Other limitations include a large attrition between the follow-up visits, lack of medication information collected at the follow-up visits, oversimplified grading of physical activity, lack of updated information about smoking status and kidney function, and lack of specific cause of death. In addition, some residual confounders, such as inflammation, infection, weight gain, pregnancy, thyroid function, and other metabolic factors, were not well adjusted in the study.

## CONCLUSIONS

This study demonstrated the importance of identifying specific heart rate trajectory patterns and elucidating their associations with all-cause mortality, coronary artery disease, and CVA in a community-based population. The markedly increased trajectory pattern was associated with a much higher risk of all-cause mortality than the stable trajectory pattern, even when the heart rate was in the normal range.

## ACKNOWLEDGEMENTS

We would like to thank the cardiologists, nurses, and medical students at National Taiwan University Hospital for their assistance with this study.

### Funding
The study was supported by the Ministry of Science and Technology, Taiwan (MOST.103-2314-B-002-135-MY3) and the National Science Council in Taiwan (NSC.102-2314-B-002-080-MY2 and NSC.100-2314-B-002-113-MY3). The funders had no role in study design, data collection and analysis, decision to publish, or preparation of the manuscript.

### Grant Disclosures
The following grant information was disclosed by the authors:
Ministry of Science and Technology, Taiwan: MOST.103-2314-B-002-135-MY3.
National Science Council in Taiwan: NSC.102-2314-B-002-080-MY2, NSC.100-2314-B-002-113-MY3.

### Competing Interests
The authors declare there are no competing interests.

### Author Contributions
- Cheng-Chun Wei conceived and designed the experiments, analyzed the data, prepared figures and/or tables, authored or reviewed drafts of the paper, and approved the final draft.
- Pei-Chun Chen analyzed the data, authored or reviewed drafts of the paper, and approved the final draft.
- Hsiu-Ching Hsu, Ta-Chen Su and Hung-Ju Lin performed the experiments, authored or reviewed drafts of the paper, and approved the final draft.

![PeerJ]

- Ming-Fong Chen and Yuan-Teh Lee conceived and designed the experiments, authored or reviewed drafts of the paper, and approved the final draft.
- Kuo-Liong Chien conceived and designed the experiments, prepared figures and/or tables, authored or reviewed drafts of the paper, and approved the final draft.

### Ethics

The following information was supplied relating to ethical approvals (i.e., approving body and any reference numbers):

The proposal has been approved by the Institutional Research Board of National Taiwan University Hospital (201003001R), and the participants' informed consent has been obtained.

### Data Availability

The raw data are available in a Supplemental File.

### Supplemental Information

Supplemental information for this article can be found online at http://dx.doi.org/10.7717/peerj.8987#supplemental-information.

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
