# Peer review of "Association of heart rate trajectories with the risk of adverse outcomes in a community-based cohort in Taiwan"

_PeerJ, doi:10.7717/peerj.8987_

## Round 0.1 · original submission · Major Revisions

Thank you for your submission which has been reviewed by 2 independent reviewers, who have found several areas of concerns. In addition, the following issues require addressing:

Critical to this work, medications data are missing.
Cause of death has not been disclosed.
It is unclear if heart rate increment is due to underlying change in status of the individual e.g. malignancy, inflammation/infection, weight gain, metabolic status (thyroid, weight change).
There is a large gap between last ECG/heart rate measures and final follow-up (> 10 years). Therefore, the trajectory is distant to outcomes?
The KM survival curves have noe included the numbers at risk over time.
How many subjects had 2 or 3 or 4 heart rate measures?
Patient characteristics change over time. Why has this not been analyzed?
Data on alcohol appears to have been collected but not presented.

·

Basic reporting

Overall, this is a well written manuscript. An appropriate background and context has been provided for this study.

Experimental design

Major issues:

• Over 14 years of follow up it is likely that many participants may be started on medication that could affect the heart rate e.g. beta blockers, rate limiting calcium channel blockers. Was this taken in to account?
• Similarly, was any consideration given to medication that could affect the heart rate at baseline e.g. inhaled corticosteroids or beta 2 agonists, anti-depressants, thyroxine?
• Did you provide any instruction to participants about use of other products that may alter heart rate e.g. caffeine and smoking, prior to heart rate assessments occurring?
• Were pregnant women included in your study cohort (as this may affect heart rate)?
• How did you treat other arrhythmias and conduction disturbances identified on ECG at baseline e.g. atrial flutter, second degree AV block, first degree AV block, frequent ectopy which may affect heart rate?
• How were individuals with arrhythmia diagnosed on ECGs during study follow up visits treated? Were they censored at this point in time?
• How did you treat individuals with pre-existing diagnoses of cardiovascular or cerebrovascular disease? Were they included in your study population? Please outline if individuals with serious comorbidity were eligible for study participation e.g. cancer?
• Can you please outline why you did not include eGFR and smoking as time updated covariates?
• In your study flow chart can you please outline reasons for reduction in the study cohort at each follow up time period? It would appear that almost two thirds of your study cohort did not attend the third follow up visit, which may impact on results obtained.
• Can you please outline how secondary outcomes were obtained? Was there a systematic search for events or was this based on participant interviews? Was there adjudication of events?
• Were all coronary heart disease outcomes required to have an angiogram for classification? If an individual had an MI without undertaking angiography, how was this classified?


Minor issues:

• In the abstract, please add in the corresponding 95% confidence intervals for the hazard ratios provided.
• In your keywords it would be worth considering the addition of ‘cerebrovascular accident’, as this was a main study outcome.
• For regular physical activity, it seems unusual that you should mandate this as two to three times per week in frequency. How were individuals who exercised with greater frequency categorised?
• You have stated that BP measurements were taken on both arms and if there was a discrepancy of >10mmHg, then another measurement was taken. How was the final BP measurement decided in these scenarios?
• Do you have any information concerning cause of death in the cohort?
• In table 4 you have listed atrial fibrillation as a covariate and, in your methodology, you have stated that this patient population was excluded – please clarify if all participants with AF were excluded or if it was only those with AF at the time of the baseline ECG.
• In Table 5 it would be clearer to state that each of the categories was compared to a stable HR trajectory in the heading or footnote as opposed to adding in ’………/stable ‘ for each trajectory, which could be misleading.

Validity of the findings

The discussion would benefit from further refining to discuss your findings and clinical implications in greater detail. It would also be useful in your discussion to refer to the approximate % change that was classified as ‘markedly increased’ to facilitate translation of your results in to clinical practice.

Additional comments

Overall, this is a clear and generally well written paper. It is lacking in methodological detail and a significant expansion of the methodology section is required.

Reviewer 2 ·

Basic reporting

The text is clear and the paper generally well-written. The Discussion could be more to the point, for example the "Main findings" repeat substantial parts of the Methods and also lists Limitations or Strengths. Some restructuring would be needed.

The paper by Seviiri et al, Heart, 2018 is important given the scarce literature in the field. It should be discussed in the Introduction and Discussion.

Tables should contain mre information to be read stand-alone, e.g. explain how P-values were calculated, or statistical models and adjustment variables etc.
Please also clarify: "based on the interquartile range of the final heart rate."

Table 1, 2, and 3 are all very descriptive only, it might be possible to merge or shorten them.

The Discussion should contain more comparison with a one-time point assessment, and possibility of bias due to selection of participants (necessary to create the trajectory). Maybe use Banack Am J Epidemiol 2019.

Tables should contain the number of participants and number of observed deaths.

Experimental design

Generally well analysed, with clear aims and appropriate methods.

It would be good to show a less adjusted model to know more about confounding. Less adjusted estimates may also be more clinically valid. I suggest adjusting only for age and sex in an additional crude analysis.

Validity of the findings

More description of the findings, including "non-significant" is needed.

Better acknowledgements of limitations and possible biases would be good too.

In this cohort, in fact, heart rate trajectories did not appear fo be very useful, did it?

The width of the confidence intervals given >500 death events were observed is quite surprising.

Additional comments

This is a generally wel-written and analysed paper. I encourage the authors to address my comments to improve their manuscript.

---

## Round 0.2 · Minor Revisions

Many thanks for the improved work however, there are residual concerns needing to be addressed. Language editing by a native English speaker is needed.

·

Basic reporting

The background is appropriate. Overall, the manuscript would benefit from English editing to improve readability.

Experimental design

There are a number of methodological areas that require addressing to improve this manuscript:
• In your study flow chart, please list exclusion reasons for all of the study cohort. Whilst deaths have been accounted for, other reasons for attrition require documenting. If individuals are lost to follow up, this should be clearly stated. As this represents approximately two thirds of your cohort, it is important that this is clearly articulated.
• I’m unclear about how you have ascertained that this attrition over time did not impact on your results in this statement: ‘The decreased number of participants in the follow-up visits did not misclassify the participants much into wrong groups and influence our results.’ Whilst you may not have misclassified the remaining participants in the cohort, it is impossible to state how the results may have been affected if there had not been such a large attrition rate.
• Please specify in the manuscript how secondary outcomes were identified, and how these events were adjudicated. This has been responded to in your initial response to my reviewer comments, but no further detail has been added into the manuscript. Furthermore, I am still unclear if systematic searching for events occurred i.e. were medical records routinely screened or did this occur only if an individual reported an event? Your methodology section should clearly state how individuals were followed up – was this by phone calls or face to face interviews? At what time intervals did follow up occur? Please also provide details for the process of adjudication of events for secondary outcomes in the manuscript.
• For coronary heart disease, please clarify if any consideration was given to angina, or those who underwent coronary angiography with documented disease but did not undergo percutaneous coronary intervention.
• Please document in the manuscript information that was given to participants to ensure that confounding factors that may affect heart rate were considered e.g. avoidance of caffeine, smoking etc and for how long prior to testing.

Validity of the findings

• In your results section, it would be useful to describe the results of the four heart rate trajectory patterns with your secondary outcome (it is in Table 5 but there is no mention of this in the text).
• In your discussion, this would benefit from explicitly stating in the opening sentences that your study demonstrated an association between heart rate trajectories and all cause mortality, but no association was evident between heart rate trajectories and cardiovascular events (secondary outcomes). It would be useful to hypothesise in your discussion about reasons underpinning this.
• I am unclear as to the meaning of this sentence in your discussion: ‘Compared with this study, we followed our participants for years after establishing the heart rate trajectory model to demonstrate that the adverse outcomes were the cause rather than the consequence of heart rate.’ Your study has not demonstrated any causation, but rather association, so I’m unsure what this means. Please clarify and re-word.

Additional comments

Overall, this is an improved version. The limitations of the study have been described but greater detail concerning methodology is required before this could be considered for publication.

Reviewer 2 ·

Basic reporting

Some text has been added to the Discussion, but the English should be improved.

Experimental design

OK

Validity of the findings

OK

Additional comments

Overall my comments were addressed quite appropriately.

Some text has been added to the Discussion, but the English should be improved.

---

## Round 0.3 · accepted · Accept

Many thanks for resubmitting the revised manuscript. The changes to the manuscript are all acceptable and I affirm that the work is now of publication standard.